# Growth Kinetics and Mechanical Properties of Rare-Earth Vanadiumizing Layer on GCr15 Steel Surface

Lingyao Meng [1], Jian Shang [1,*], Mengjiu Zhang [2], Aijun Xie [2] and Yue Zhang [1]

[1] School of Materials Science and Engineering, Liaoning University of Technology, No. 169 Shiying Street, Guta District, Jinzhou 121000, China; mly117589@163.com (L.M.); 348349585@163.com (Y.Z.)
[2] Qingdao Choho Industrial Co., Ltd., No. 112 Hongkong Road, Qingdao 266705, China; mzhy236@163.com (M.Z.); xieaijun08@163.com (A.X.)
* Correspondence: shangbahao@163.com

**Abstract:** In this paper, rare-earth vanadiumizing layers were prepared on the surface of GCr15 steel by powder pack cementation. The tissue thicknesses of the vanadiumizing layers were characterized by metallographic microscopy, scanning electron microscopy, X-ray diffraction, electron microprobe analysis and microhardness testing at 1173, 1193, 1213 and 1223 K for 1, 3, 5 and 7 h, respectively, and the growth kinetics of the rare-earth vanadiumizing layers were investigated first. The experimental results showed that: a dense and uniform vanadiumizing layer was obtained on the surface of the substrate, and the layer mainly consisted of $VC_x$ and α-Fe; the thickness of the vanadiumizing layer increased with the increase in heating temperature and holding time, and the variation range was 4.65–12.65 μm; the microhardness of the vanadiumizing layer increased with the increase in heating temperature and holding time, and the variation range was 1892.3–2698.6 HV, compared with the substrate. The electron probe microanalysis showed that the rare earth entered the diffusion layer and affected its tissue hardness. The experimental diffusion activation energy of the GCr15 steel powder-embedded rare-earth vanadiumizing layer was 164.85 KJ/mol.

**Keywords:** GCr15 steel; vanadiumizing; powder pack cementation; rare earth





## 1. Preface

GCr15 steel is widely used in bearings [1], die tools [2] and transmission chains [3]. For transmission chains, axis pins are prone to surface failure due to wear, fracture and fatigue during chain transmission, so the surface strengthening treatment of axis pins to meet their increasing technical requirements for hardness and wear resistance is an effective way to improve the service life of transmission chains [4,5]. Metal carbide cladding, such as chromium carbide, vanadium carbide and niobium carbide, with high hardness and a low coefficient of friction, has been applied for the surface strengthening of die steel [6–12]. Researchers mostly use the thermal reactive diffusion method (TRD) to obtain metal carbide layers, which can be broadly classified into two types, the salt bath method and the solid method, depending on the treatment method. Compared to the solid method, the adhesion of the diffusion layer surface is greater at the end of the test in the salt bath method, which is more difficult to clean, and the short crucible life costs a lot of money, which is not suitable for smaller-size parts such as axis pins.

TRD is a diffusion process that requires higher temperature and longer holding conditions to obtain carbide cladding, and too high a temperature and too long a time will cause decreasing denseness and increasing brittleness of the diffusion layer, increasing the deformation of the workpiece and the coarsening of the matrix microstructure and grains; therefore, researchers have carried out research on rare-earth-modified carbide cladding. Studies have shown that the addition of rare earths in carburizing, nitriding and boriding can effectively improve the diffusion rate, hardness and mechanical properties of the diffusion layer [13–16]. After adding small amounts of rare-earth elements to the

nitriding process, it was found that the addition of rare earths significantly increased the surface hardness and hardening depth of the nitriding layer and significantly improved the vein-like microstructure of the surface diffusion layer [14]. By adding different contents of $CeO_2$ to a boron cementation agent, Fang found that the addition of rare earths has a critical value, and excessive addition will inhibit boron penetration [15]. Tao studied the microstructure and mechanical properties of a boron vanadium co-diffusion layer by doping the powder with the rare-earth compound $CeCl_3$ and found that the addition of rare earth increased the permeation rate by 40% and greatly improved the hardness and wear resistance of the layer [16]. The effect of rare earths on the microstructure and properties of vanadium carbide coatings was investigated by adding different contents of FeSiRe23 to a salt bath, and it was found that the addition of rare earths promoted the growth rate of the coatings, reduced the grain size of the coatings and improved the brittleness of the coatings [17].

In summary, the effects of rare earths on the microstructure and properties of metal diffusion layers are mostly concentrated in the salt bath infiltration system, and little research has been reported on the powder pack cementation of rare-earth metal diffusion layers. After powder pack cementation treatment, the surface of the workpiece adheres to fewer powder particles, which are easy to remove and convenient for subsequent processing, suitable for the surface treatment of small-sized, high-volume axis pin parts. Therefore, this study adopted the powder pack cementation method for the vanadiumizing treatment of GCr15 steel, investigated the effect of the heating temperature and holding time on the microstructure and properties of the vanadiumizing layer of GCr15 steel, and analyzed and calculated the growth of the rare-earth diffusion layer on the mechanics so as to provide a reference for the application of rare-earth diffusion layers in pin shafts, among others.

## 2. Experimental Materials and Methods

### 2.1. Experimental Materials and Vanadiumizing Process

Table 1 shows the chemical composition of GCr15 steel. The experimental samples were $\Phi 20 \times 4$ mm in size. Normalization was used as a pre-treatment, and the microstructure is shown in Figure 1. After normalizing, the samples were polished with $120^{\#}$–$1000^{\#}$ sandpaper until the surface was bright and free of scratches, and the samples were ultrasonically cleaned with anhydrous ethanol and blown dry prior to the vanadiumizing process. The vanadium supply agent (ferro-vanadium powder), activated agent ($NH_4Cl$), catalyst ($La_2O_3$) and filler agent ($Al_2O_3$) were dried separately, and then 47% V-Fe + 5% $NH_4Cl$ + 1% $La_2O_3$ + 47% $Al_2O_3$ was weighed, mixed, and added to a stainless-steel container. When the samples were packed, a mixture of water glass ($Na_2O$-$SiO_2$) and refractory clay was used to seal the container. The treatment schematic diagram is shown in Figure 2. The sealed container was put into a Muffle furnace and held for 1, 3, 5 and 7 h at 1173, 1193, 1213 and 1233 K with different rare-earth additions, respectively.

After the vanadiumizing treatment, the stainless-steel container with samples was air-cooled to room temperature. The removed samples were first analyzed using a D/MAX-2500 X-ray diffractometer from Rigaku, Tokyo, Japan. The scanning speed was $8°$/min, and the scanning angle was $30°$–$90°$. The samples were mounted with resin and polished with sandpaper, and they were then corroded with 4% nitric acid alcohol to observe the sectional microstructure and analyze the composition of the prepared layer. The Axio Vert. A1 metallographic microscope produced by Zeiss (Berlin, Germany, A Sigma-500 SEM) with an energy spectrometer from Oxford, UK, was used to characterize the layer's sectional microstructure and to determine the element distribution. The hardness of the prepared layer was tested 7 times using the HM-200 microhardness tester from Aoshengtai, Qingdao, China, under a load of 0.2 kg and with a 10 s holding duration. The phase temperature of mixed powder was tested using a US-PerkinElmer-STA6000 differential thermal analyzer (PerkinElmer, Washington, DC, USA) in the range of 0–980 °C and at a rate of 10 °C/min for 1.5–2 h. The Japan-JEOL-JXA 8530F electronic probe (JEOL, Tokyo, Japan) was used to

detect the rare-earth distribution in the prepared layer and ensure that the rare earths were involved in the layer formation process.

**Table 1.** Chemical composition of GCr15 steel.

| Chemical Element | C | Mn | Si | S | Cr | P | Ni |
|---|---|---|---|---|---|---|---|
| Mass fraction (wt.%) | 0.95–1.05 | 0.25–0.45 | 0.15–0.35 | ≤0.020 | 1.40–1.65 | ≤0.027 | ≤0.30 |

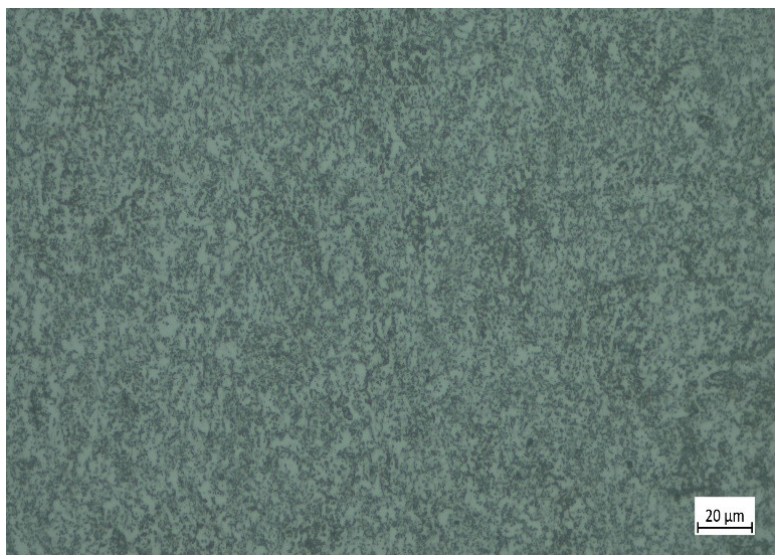

**Figure 1.** Microstructure of normalized GCr15 steel.

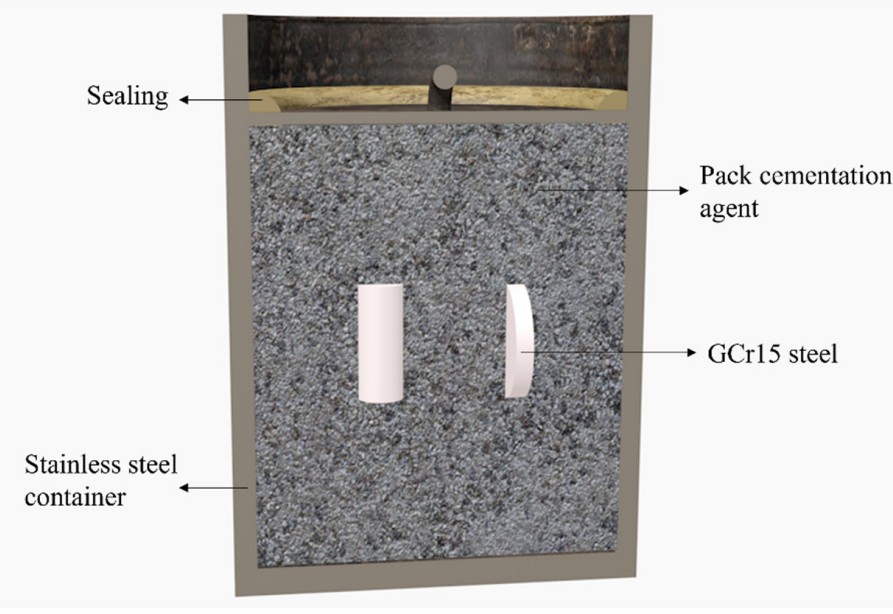

**Figure 2.** Vanadiumizing treatment schematic diagram.

## 2.2. Kinetic Analysis Method

Some scholars have studied the growth kinetics of carbide coatings in the TRD process [18,19], especially in salt bath conditions. The results show that the formation and growth of carbide coatings in the TRD process are mainly controlled by the diffusion law, so the growth kinetics of rare-earth vanadiumizing layers can be described by the classical

parabolic growth law, where the growth rate of the vanadiumizing layer is parabolically related to the holding time during the vanadiumizing process, satisfying:

$$d^2 = Dt \tag{1}$$

where d represents the permeate thickness, mm; t represents the diffusion time, s; and D represents the atomic diffusion coefficient, $cm^2/s$. The relationship between the diffusion coefficient D and the diffusion activation energy Q and treatment temperature T can be expressed by the Arrhenius equation:

$$D = D_0 \exp(-Q/RT) \tag{2}$$

where $D_0$ is the constant, $cm^2/s$; R is the gas constant (8.314 J/mol·K).

## 3. Experimental Results and Analysis

### 3.1. Diffusion Layer Cross-Sectional Microstructure and Composition

Figure 3 shows the metallographic microstructures of the rare-earth vanadiumizing layers obtained after 1 and 7 h of vanadiumizing treatment at 1173, 1193, 1213 and 1233 K. It can be seen in Figure 3 that after the vanadiumizing treatment, the surface of the substrate formed a bright white layer with different thicknesses; under different temperature conditions, the thickness of the vanadiumizing layer was uniform, and there was no apparent transition zone between the layer and the substrate. As the heating temperature increased and the holding time grew, the vanadiumizing layer also showed a gradual thickening trend. Figure 4 shows the cross-sectional SEM images and energy spectrum distribution of Figure 3c. In Figure 4, it can be seen that the vanadiumizing layer can be divided into three parts: (1) the surface diffusion zone, (2) the transition zone between the matrix and the diffusion layer and (3) the internal matrix zone. The vanadium element is mainly distributed in the surface diffusion zone, the iron element is mainly distributed in the internal matrix zone, and the carbon element is distributed throughout the matrix and diffusion layer and shows enrichment in the diffusion layer, which is because the solubility of carbon in the diffusion layer is greater than that of the matrix, the migration of carbon atoms to the diffusion layer occurs, and the carbon content tends to increase from the matrix to the surface. Meanwhile, the presence of rare earths was not found after the surface scan. On the one hand, it is assumed that rare earths are involved in the reaction, but there is no visible signal in the surface scan due to the small amount added; on the other hand, it is assumed that rare earths are not involved in the reaction, and rare earths remain in the remaining diffusion agent, so there is no signal in the surface scan.

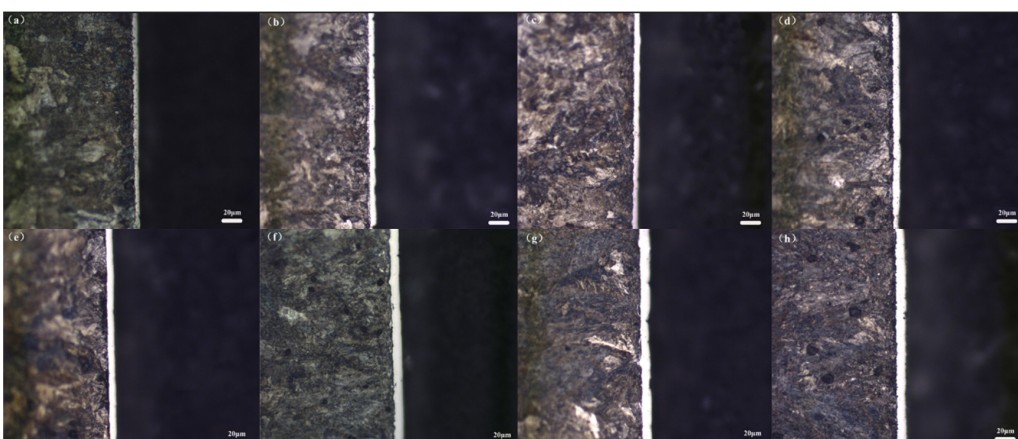

**Figure 3.** Sectional microstructure of the vanadiumizing layer prepared at different heating temperatures for 1 h and 7 h. (**a**) 1173 K 1 h; (**b**) 1193 K 1 h; (**c**) 1213 K 1 h; (**d**) 1233 K 1 h; (**e**) 1173 K 7 h; (**f**) 1193 K 7 h; (**g**) 1213 K 7 h; (**h**) 1233 K 7 h.

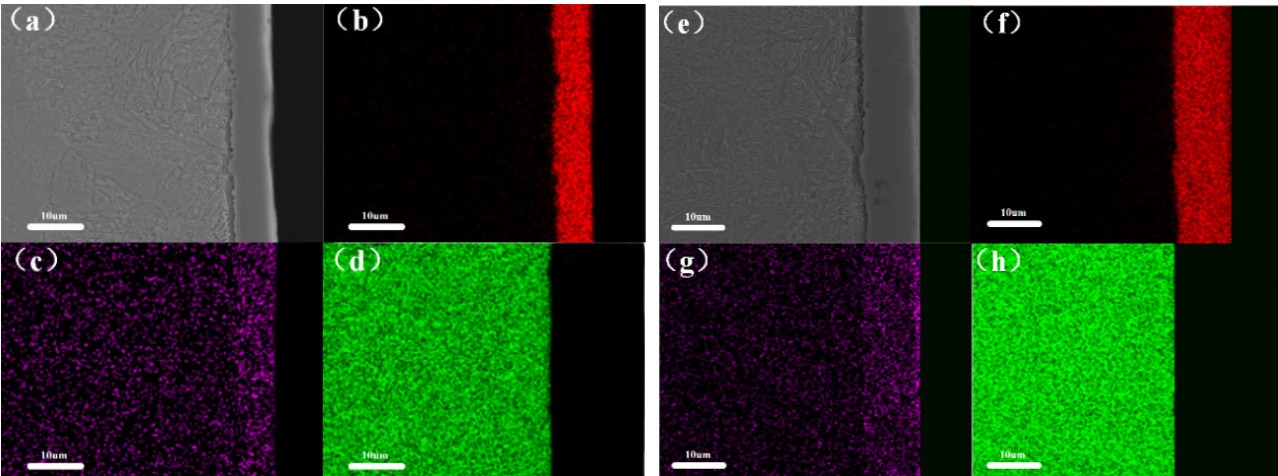

**Figure 4.** SEM morphology and EDS analysis of the vanadiumizing layer. (**a**,**e**) SEM image of vanadiumizing layer; (**b**,**f**) distribution of vanadium element; (**c**,**g**) distribution of carbon element; (**d**,**h**) distribution of iron element.

The differential thermal analysis curves of the heating process with and without the rare-earth $La_2O_3$ diffusion agent are shown in Figure 5, in which it can be seen that there are three heat absorption peaks in the continuous heating process of the diffusion agent. The Gibbs–Helmholtz equation and the Klebron equation can be used to calculate the Gibbs free energy $\Delta G$ versus temperature for the reactions to produce VC and $V_2C$, as shown in Figure 6. It can be seen that the Gibbs free energy for the generation of $V_2C$ is much smaller than that for the generation of VC, and thus, there is a greater tendency to generate $V_2C$ in the early stages of the reaction, while it can be deduced from the C-V phase diagram that heat absorption peak 1 represents the formation of $\alpha$-$V_2C$; heat absorption peak 2 represents the generation of $VC_x$; and heat absorption peak 3 corresponds to the formation of $\beta$-$V_2C$. However, the above heat absorption peaks are not in absolute correspondence with solid precipitation, and $VC_x$ is also generated when $\alpha$-$V_2C$ and $\beta$-$V_2C$ are generated. Comparing the curves, it can be seen that the area of the three heat absorption peaks is significantly larger after the addition of rare-earth $La_2O_3$ than that without $La_2O_3$, which indicates that the addition of rare earth does promote the formation of the diffusion layer. Figure 7 shows the electron probe trace analysis of the rare-earth vanadiumizing layer: Figure 7a shows the La element content without the addition of rare earth, and Figure 7b shows the La element content with the addition of 1% rare earth. The comparison in Figure 7 shows that the rare-earth La element signal ranges from 2.3–3.5 when the rare-earth addition is 0% and from 2.7–4.9 when the rare earth addition is 1%. The comparison reveals that the rare-earth elements are indeed involved in the reaction, confirming the above hypothesis. Figure 8 shows the X-ray diffraction spectra of the surfaces of samples treated by vanadiumizing at 1173, 1193, 1213 and 1233 K for 1 and 7 h. The results show that the surface of the diffusion layer is mainly composed of $VC_x$ and $\alpha$-Fe, but with the growth of heating temperature and holding time, the intensity of the $\alpha$-Fe phase gradually decreases, and after the temperature reaches 1213 K for 7 h, the $\alpha$-Fe phase disappears, which is because of the local diffusion layer thinness. The GCr15 steel matrix was detected by ray penetration of the diffusion layer; the $VC_x$ grains have a meritocratic orientation in the (111) and (200) grain planes, while no rare-earth compounds were found on the diffusion layer surface.

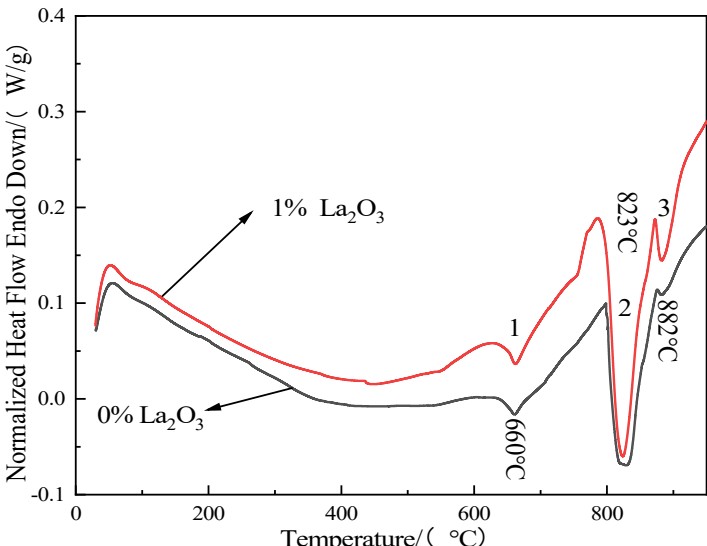

**Figure 5.** Differential thermal analysis curves with and without rare-earth La$_2$O$_3$ in powder.

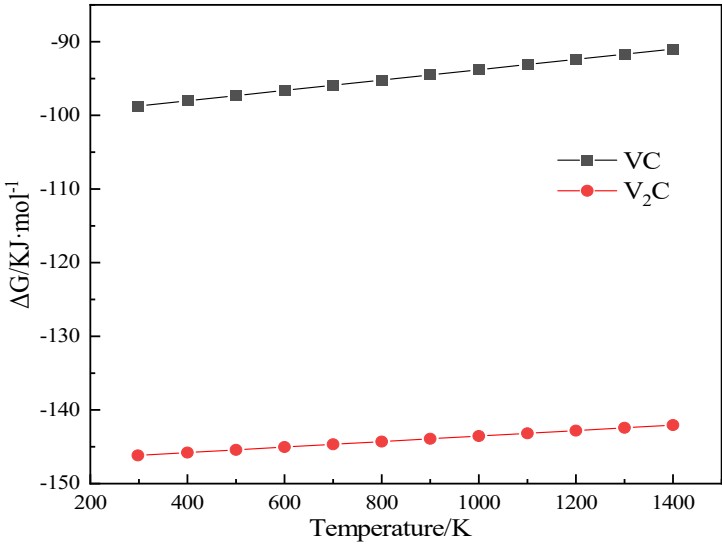

**Figure 6.** Temperature dependence of ΔG of VC and V$_2$C during V-C reaction.

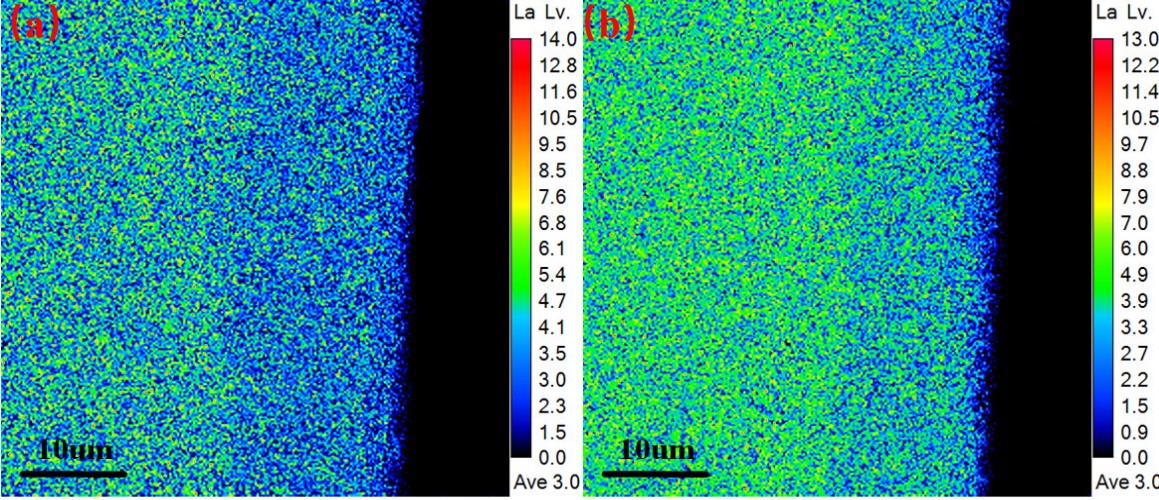

**Figure 7.** EPMA analysis of La element in vanadiumizing layer: (**a**) 0% La$_2$O$_3$ and (**b**) 1% La$_2$O$_3$.

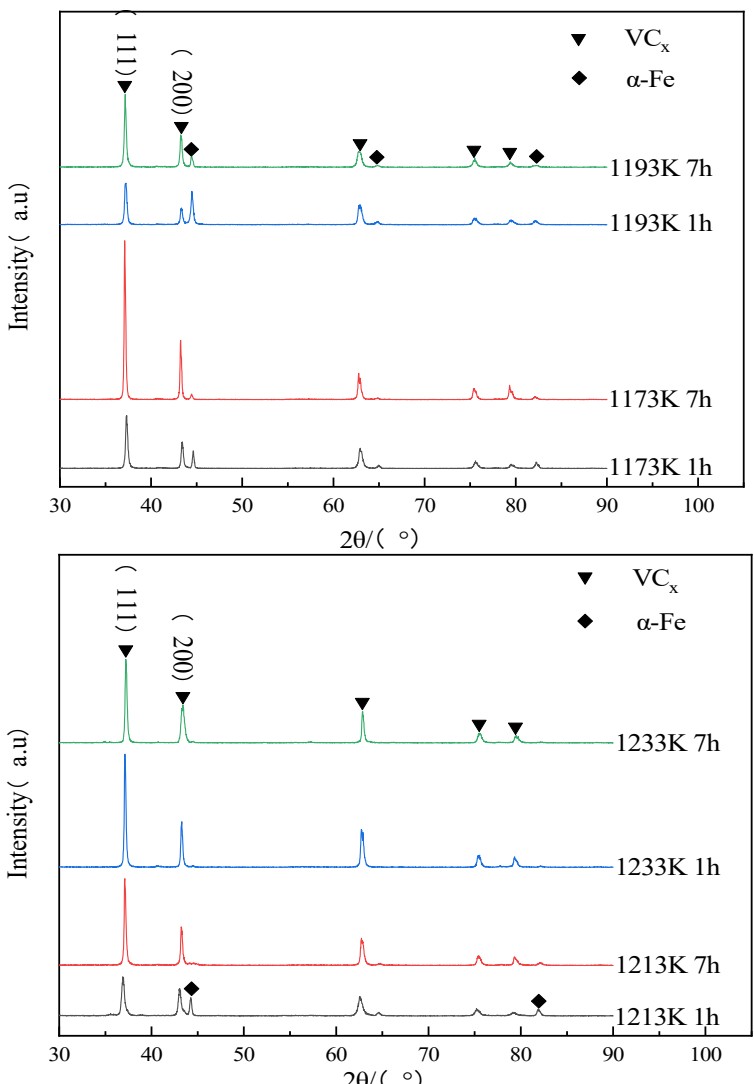

**Figure 8.** X-ray diffraction spectrum of vanadiumizing layer prepared at different heating temperatures for 1 and 7 h.

### 3.2. Thickness and Hardness of Diffusion Layer

The graphs of rare-earth vanadiumizing layer thickness versus holding time obtained at 1173, 1193, 1213 and 1233 K heating conditions for 1, 3, 5 and 7 h are shown in Figure 9. In Figure 9, it can be seen that the thickness of the vanadiumizing layer varies from 4.65–12.65 μm with the treatment temperature and holding time. At 1173 K, the thickness of the vanadiumizing layer varies from 4.65–8.85 μm; at 1193 K, the thickness of the vanadiumizing layer varies from 6.08–11.61 μm; at 1213 K, the thickness of the vanadiumizing layer varies from 7.08–12.22 μm; at 1233 K, the thickness of the vanadiumizing layer varies from 7.88–12.65 μm. This shows that the thickness of the vanadiumizing layer does not vary significantly between 1213–1233 K. The thickness of the rare-earth vanadiumizing layer increases with the increase in the holding time at different treatment heating temperatures. For the same holding time, the rare-earth vanadiumizing layer thickens substantially with increasing heating temperature.

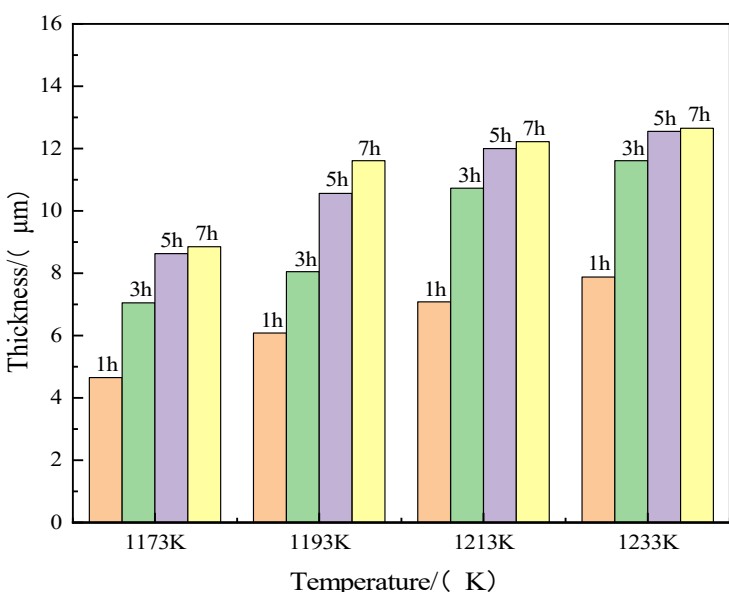

**Figure 9.** The thickness variation in vanadiumizing layer prepared at different heating temperatures and durations.

Figure 10 shows the hardness variation in the rare-earth vanadiumizing layer at different heating temperatures and different holding times. As can be seen in Figure 10, the Vickers hardness of the vanadiumizing layer varies from 1892.3–2698.6 HV depending on the treatment temperature and holding time, which is the same trend as the thickness of the vanadiumizing layer, and the hardness of the rare-earth vanadiumizing layer increases with the increase in thermal temperature and holding time. Compared to the base material (230 HV), the hardness is increased by a factor of about 11–12.

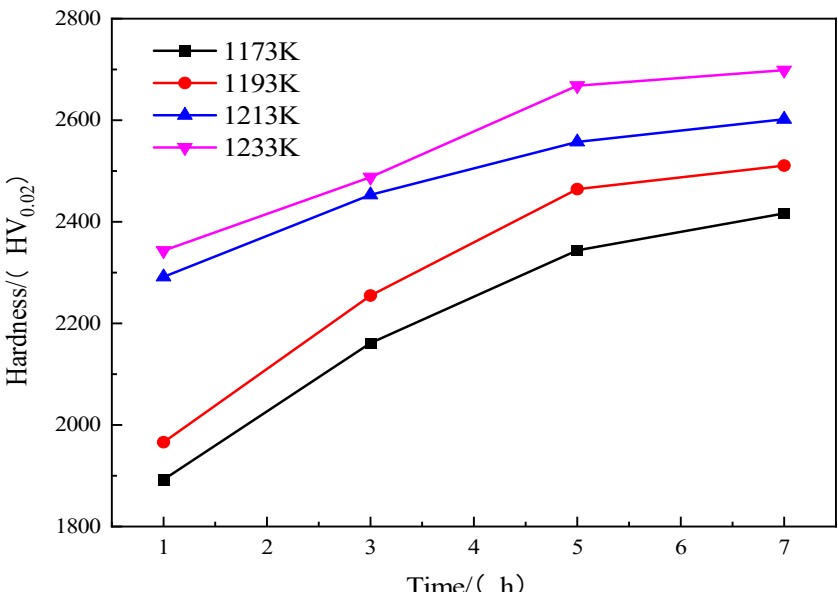

**Figure 10.** The hardness variation of vanadiumizing layer prepared at different heating temperatures and durations.

### 3.3. Growth Kinetics of Diffusion Layers

Figure 11 shows the fitted curves of the square of the thickness of the rare-earth vanadiumizing layer at different heating temperatures (1173, 1193, 1213 and 1233 K) versus the holding time (1, 3 and 5 h). It can be seen in Figure 11 that the square of the thickness of

the vanadiumizing layer is linearly related to the vanadiumizing time, and with the linear fitting of the heating temperatures at 1173, 1193, 1213 and 1233 K versus the holding time, the resulting slope of the straight line is the atomic diffusion coefficient D of the rare-earth vanadiumizing layer.

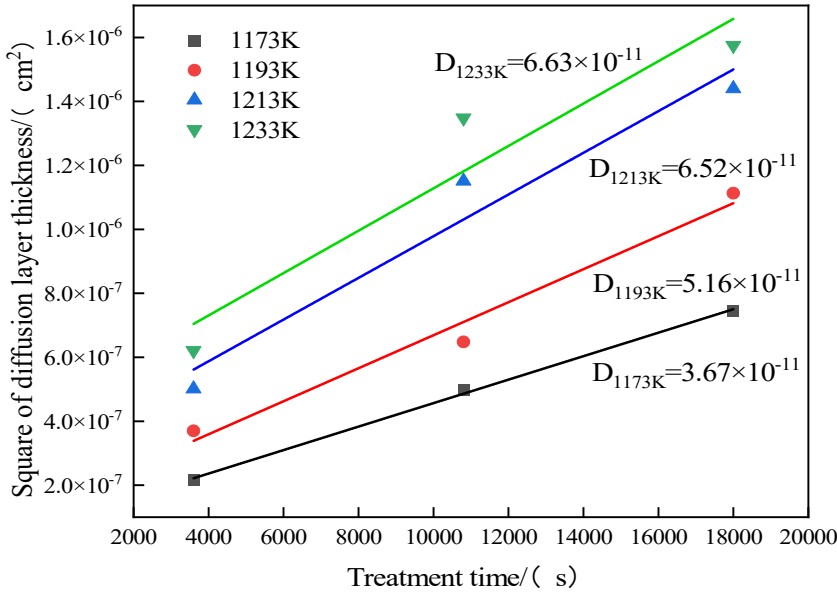

**Figure 11.** Square of vanadiumizing layer thickness versus duration at different temperatures.

Taking the logarithm of both sides of Equation (2), as shown in Equation (3), it can be found that lnD is linearly related to 1/T. The relationship between the growth rate of the rare-earth vanadiumizing layer and the heating temperature can be obtained by linearly fitting lnD to 1/T at 1173, 1193, 1213 and 1233 K, as shown in Figure 12. The slope of the fitted line is the diffusion activation energy Q = 164.85 KJ/mol for the vanadiumizing layer when rare earths are added.

$$lnD = lnD_0 - Q/RT \tag{3}$$

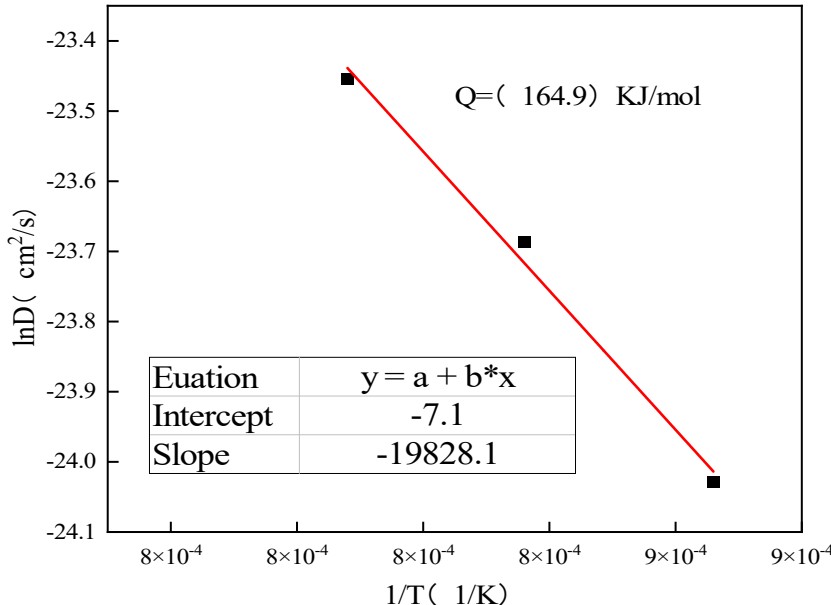

**Figure 12.** Fitted graph of the variation of lnD versus 1/T.

The growth kinetics of vanadium carbide coatings on several substrates applied by the TRD technique was determined according to Equation (1), and the results of $D_0$ and $Q$, calculated according to Equation (2), are summarized in Table 2. In Table 2, it can be seen that the growth rate constant of the vanadiumizing layer shows great differences for different substrates, and the diffusion activation energy of the vanadiumizing layer prepared by different methods also has great differences, which indicates that the thickness of the vanadiumizing layer depends not only on the heating temperature and holding time but also on the selection of the vanadiumizing method and substrate.

**Table 2.** $D_0$ and $Q$ of vanadiumizing layer under different substrate conditions.

| Substrate | $D_0/(\times 10^{-4}\ cm^2/s)$ | $Q/(KJ/mol)$ | Method | Ref. |
|---|---|---|---|---|
| DIN 1.2367 | 7.98 | 173.2 | Salt bath | [20] |
| D2 | 25.80 | 123.3 | Salt bath | [21] |
| Ck45 | 0.13 | 186.7 | Salt bath | [22] |
| C 105 W1 | 0.16 | 170.0 | Salt bath | [22] |
| GCr15 | 0.02 | 187.3 | Salt bath | [23] |
| GCr15 | 0.12 | 164.9 | Solid powder | Present study |

## 4. Conclusions

1. A dense and uniform vanadiumizing layer was obtained on the surface of GCr15 steel. There was no visible transition zone between the vanadiumizing layer and the matrix. X-ray diffraction analysis showed that the diffusion layer was mainly composed of $VC_x$ and $\alpha$-Fe and had a preferred orientation in the (111) and (200) crystal planes. With the increase in heating temperature, the thickness of the diffusion layer increased, and the $\alpha$-Fe phase gradually disappeared.

2. The thickness and microhardness of the vanadiumizing layer increased with heating temperature and holding time; the thickness and hardness of the layer were 4.65–12.65 μm and 1892.3–2698.6 $HV_{0.02}$, respectively.

3. Kinetic calculations revealed that the diffusion activation energy of the vanadiumizing layer was 164.85 KJ/mol under rare-earth catalytic conditions, and the combination of differential thermal analysis and EPMA results revealed that rare earths were involved in the vanadiumizing reaction and contributed to the formation of the vanadiumizing layer.

**Author Contributions:** Conceptualization, L.M. and J.S.; methodology, J.S.; software, L.M.; validation, L.M., J.S., A.X., Y.Z. and M.Z.; formal analysis, J.S.; investigation, J.S.; resources, M.Z.; data curation, L.M.; writing—original draft preparation, L.M.; writing—review and editing, L.M.; supervision, J.S.; project administration, J.S. All authors have read and agreed to the published version of the manuscript.

**Funding:** This research received no external funding.

**Institutional Review Board Statement:** Not applicable.

**Informed Consent Statement:** Not applicable.

**Data Availability Statement:** Not applicable.

**Conflicts of Interest:** The authors declare no conflict of interest.

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
