# Peer review of "Growth Kinetics and Mechanical Properties of Rare-Earth Vanadiumizing Layer on GCr15 Steel Surface"

_coatings, doi:10.3390/coatings12071018_

Round 1

Reviewer 1 Report

There are too much "on the other hand" in one sentence, on 4th page.

Reviewer 2 Report

The idea of ​​using vanadization in a powder mixture with rare earth elements in TRD is technically interesting and therefore any information about their influence on the coating process is valuable. Unfortunately, the authors did not use all the possibilities provided by the used experimental techniques when writing the article, and therefore some conclusions are not sufficiently demonstrable.

I have the following critical comments about the work.

1.     The description of the experiments is too general. More detailed data on measurement conditions as well as equipment manufacturers need to be expanded and added (type of microanalyzer used and measurement conditions are not specified, U and I are not specified for diffraction measurements, such as diffractometer method, geometry and optics, SEM and EDX analysis are also missing information on measurement conditions, the used thermal analyzer and measurement conditions are not specified, ...)

2.     Metallographic analysis refers to the Chinese standard JB / T 5069-2007. The article is intended for readers from other regions, so it is necessary to provide essential information about the preparation of samples, as well as used etchants.

3.     The method of kinetic analysis is very general - to extend.

4.     The scales in FIG. 2 are small and insufficiently legible. Image quality is very poor (blurred on the left). I request to supplement the initial microstructure of the used steel after spheroidization annealing.

5.     In FIG. 3 does not show the display mode (SEI / BEI). I assume that the SEI mode has been used, but due to the layer composition it is necessary to use the BEI mode. The basic problem of TRD in vanadization is the formation of a double layer MCx / M2C, which depends on the process parameters. For EDX analysis, I would therefore prefer line analysis instead of mapping. It needs to be supplemented.

6.     Conclusions about the role of rare earth elements only on the basis of the area distribution of the elements are inadequate. This can only be stated on the basis of quantitative measurements, which, however, are not mentioned in the article.

7.     Who performed the thermodynamic calculations in FIG. 5? Are these your own calculations? Then the calculation methodology must be stated in the description of the experiments.

8.     In FIG. 6a is the mapping of La in the sample without the addition of La2O3, will the authors of the article explain its presence in the layer and diffusion zone?

9.     FIG. 7. It is not clear which diffraction analysis methodology was used. Has the grazing incidence method been used? Based on which database the individual peaks were assigned to the phases. The phases must be clearly identified according to a PDF database or similar. Is it really a monolayer? On the diffraction pattern at 1233K / 1h is a weak peak of probably hexagonal phase V2C.

10.  The conclusions are only that the results have been achieved. There is a lack of serious discussion about the achieved results. An analysis of the effect of high vanadization temperature on the microstructure of GCr15 steel and subsequent heat treatment procedures was not performed.

11.  The references lack authors from countries other than China, who have been dealing with this issue for a long time.

Reviewer 3 Report

The material presented in the article has applied value. The research methodology and discussion of the results obtained do not contradict the basic approaches of the Science of Materials. However, I would like to draw the authors' attention to the following remarks.

1.     A fairly broad problem on surface engineering methods in Section " Preface " is not fully covered, which indicates a relatively small number of citations.

2.     Section 1.2 does not carry new information, the above formulas are known from the educational literature, so I think that it could have been deleted and limited to a reference.

3.     When discussing the results of the study, the authors found no rare earth elements or secondary phases in the surface layers. This is quite expected, given their ability to evaporate with increasing temperature. This should have been mentioned. In this case, one can look more optimistically at the results presented in Fig. 6 and 7.

4.     Conclusion No. 1 is obvious without carrying out experiments due to the high affinity of vanadium for carbon, the amount of which is large in die steel (Table 1). At the same time, Conclusion No. 2 is insufficiently substantiated. It is quite clear that the depth of the synthesized coating will increase with an increase in temperature and isothermal holding time. As for the increase in microhardness, this must be substantiated.

Maybe these articles will allow you to look more sharply at the Problem that you are solving.

https://doi.org/10.1051/epjap/2012110002-

https://doi.org/10.3390/ma15082707
